# Tumor Sink Effect with Prostate-Specific Membrane Antigen-Targeted Theranostics in Patients with Metastatic Castration-Resistant Prostate Cancer: Intra-Individual Evaluations

**DOI:** 10.3390/cancers15092592

**Published:** 2023-05-03

**Authors:** Caroline Burgard, Florian Rosar, Robert J. Marlowe, Mark Bartholomä, Sebastian Dewes, Andrea Schaefer-Schuler, Johannes Linxweiler, Fadi Khreish, Samer Ezziddin

**Affiliations:** 1Department of Nuclear Medicine, Saarland University—Medical Center, 66421 Homburg, Germany; caroline.burgard@uks.eu (C.B.); florian.rosar@uks.eu (F.R.); mark.bartholomae@uks.eu (M.B.); nuklearmedizin@uks.eu (S.D.); andrea.schaefer@uks.eu (A.S.-S.); fadi.khreish@uks.eu (F.K.); 2Spencer-Fontayne Corporation, Jersey City, NJ 07304, USA; rjm031964@cs.com; 3Department of Urology, Saarland University—Medical Center, 66421 Homburg, Germany; johannes.linxweiler@uks.eu

**Keywords:** tumor sink effect, prostate-specific membrane antigen (PSMA), positron emission tomography/computed tomography (PET/CT), radioligand therapy (RLT), metastatic castration-resistant prostate cancer (mCRPC)

## Abstract

**Simple Summary:**

The toxicity and dosing of radioligand therapy (RLT) of metastatic castration-resistant prostate cancer (mCRPC) may be impacted by the “tumor sink effect”. This phenomenon occurs when a bulky tumor accumulates so much radiopharmaceutical that uptake of the agent significantly decreases in healthy organs-at-risk of radiation-related damage. We assessed the tumor sink effects of prostate-specific membrane antigen (PSMA)-targeted radiopharmaceuticals via three within-patient comparisons in 33 men with mCRPC given two cycles of lutetium-177-PSMA-617 RLT. The comparisons involved changes in the relationships between total lesion PSMA, reflecting the body’s overall tumor burden, and/or the mean standardized uptake value, reflecting radiopharmaceutical accumulation, in the parotid salivary glands, spleen, liver, and kidney. Tumor sink effects were found in the salivary glands and spleen, and possibly the liver. These findings support everyday use and additional study of individualized lutetium-177-PSMA-617 activities adjusted based on tumor burden changes over the course of RLT in men with advanced prostate cancer, to try to improve the efficacy/toxicity ratio.

**Abstract:**

“Tumor sink effects”, decreased physiological uptake of radiopharmaceuticals due to sequestration by a tumor, may impact radioligand therapy (RLT) toxicity and dosing. We investigated these effects with prostate-specific membrane antigen (PSMA)-targeted radiopharmaceuticals in the healthy organs-at-risk (the parotid glands, kidneys, liver, and spleen) of 33 patients with metastatic castration-resistant prostate cancer (mCRPC). We retrospectively performed three intra-individual comparisons. First, we correlated changes from baseline to post-RLT (after two 177-lutetium (177Lu)-PSMA-617 cycles) in total lesional PSMA (∆TLP) and organ mean standardized uptake values (∆SUVmean). Second, in 25 RLT responders, we compared the organ SUVmean post-RLT versus that at baseline. Lastly, we correlated the baseline TLP and organ SUVmean. Data were acquired via 68-gallium-PSMA-11 positron emission tomography before the first and after the second 177Lu-PSMA-617 cycle. In the parotid glands and spleen, ∆TLP and ∆SUVmean showed a significant inverse correlation (r = −0.40, *p* = 0.023 and r = −0.36, *p* = 0.042, respectively). Additionally, in those tissues, the median organ SUVmean rose significantly from baseline after the response to RLT (*p* ≤ 0.022), and the baseline TLP and SUVmean were significantly negatively correlated (r = −0.44, *p* = 0.01 and r = −0.42, *p* = 0.016, respectively). These observations suggest tumor sink effects with PSMA-targeted radiopharmaceuticals in the salivary glands and spleen of patients with mCRPC.

## 1. Introduction

The term “tumor sink effect” refers to decreased physiological uptake of a radiopharmaceutical due to abundant sequestration of that agent by a ligand receptor expressed on malignant tissue [1,2]. Put simplistically, the phenomenon entails a large volume of tumor “absorbing” so much radiopharmaceutical that only a reduced amount of that agent, and therefore radiation, reaches healthy tissues. The clinical relevance of the tumor sink effect thus lies in the phenomenon’s potential impact on the off-target toxicity of radioligand therapy (RLT) for cancer, and hence, the dosing of the radiopharmaceutical [3]. A tumor sink effect could permit safe and tolerable administration of larger, possibly more efficacious RLT activities in patients with high tumor burdens. In these individuals, the bulky tumor would “shield” the healthy tissues from much of the additional radiation exposure from the increased amount of radiopharmaceutical. Conversely, the phenomenon could heighten the risk of adverse events with conventional RLT activities in patients with low tumor burdens. In such patients, the reduced volumes of target tissue would take up a limited amount of the radiopharmaceutical and hence increase the exposure of healthy tissues to radiation from even conventional dosing of that agent.

The first documentation of the tumor sink effect was in the RLT of a neuroendocrine tumor [4]. The phenomenon has also been speculated to be “uncommon but possible” in radioiodine thyroid remnant ablation and radioiodine treatment of differentiated thyroid carcinoma metastases [5].

In the last decade, RLT of prostate cancer with radiopharmaceuticals targeted at prostate-specific membrane antigen (PSMA) has gained increasing clinical acceptance and more recently, regulatory approval to treat advanced prostate cancer. In this setting, such therapy frequently exerts potent anti-tumor effects against established lesions, although its effects in preventing metastatic spread remain to be better characterized. The clinical acceptance and regulatory approval have come because PSMA-targeted RLT has been shown to confer survival and palliative benefits, while causing generally limited toxicity, in patients with metastatic prostate cancer, even those who are very frail and/or have very-late-stage disease [6,7,8,9,10,11,12,13]. 

For this reason, the question of whether, where, and to what extent a tumor sink effect exists with PSMA-targeted radiopharmaceuticals in men with prostate cancer is of rising clinical relevance. Thus to our knowledge, at least seven published clinical studies [2,14,15,16,17,18,19], one published simulation study [1], and one case report [20] have sought to address this issue. These investigations had disparate cohort characteristics and methodology, and contradictory results. 

We therefore sought to further characterize tumor sink effects, if any, in key healthy organs-at-risk of patients with metastatic castration-resistant prostate cancer (mCRPC) given two courses of 177-lutetium (^177^Lu)-PSMA-617 RLT. To do so, for each organ, we performed three types of intra-individual comparison of PSMA radioligand uptake. We also sought to determine via statistical analysis whether key demographic, clinical, biochemical, molecular, or therapeutic factors could predict any tumor sink effect in each of two of the studied organs-at-risk. 

## 2. Patients and Methods

### 2.1. Study Design and Endpoints

This was a retrospective analysis of patients in a prospective “real-world” registry study: Prospective **RE**gistry to **A**ssess Outcome and Toxicity of Targeted Radionuc**LI**de **T**herap**Y** in Patients with mCRPC in Clinical Routine (REALITY Registry; clinicaltrials.gov identifier NCT04833517) [9]. 

We sought to evaluate the magnitude of tumor sink effects, if any, in each of the four key healthy organs-at-risk of RLT toxicity [11,21,22,23,24], namely, the salivary glands (right and left parotid glands), spleen, kidneys, and liver of patients with mCRPC. Because the results for the right versus the left parotid gland and the right versus the left kidney did not differ significantly, we report data for the right parotid gland and right kidney for simplicity’s sake. The presence of tumor sink effects was assessed both before (“baseline”) and after two cycles of ^177^Lu-PSMA-617 monotherapy (“post-RLT”).

The first endpoint comprised the correlations between changes from baseline to post-RLT in total lesion PSMA (∆TLP), a measure of aggregate tumor burden, and changes in the mean standardized uptake values (∆SUVmean) in each respective organ. The second endpoint was changes in organ SUVmean from baseline to post-RLT in molecular imaging responders to RLT. The third endpoint was the correlation of baseline TLP and baseline organ SUVmean in the overall study sample. For all comparisons, TLP and SUVmean were calculated based on data from routine serial 68-gallium (^68^Ga)-PSMA-11 positron emission tomography/computed tomography (PET/CT), i.e., a pre-RLT scan to screen for eligibility for ^177^Lu-PSMA-617 and a second scan to assess the response to the first two RLT courses.

### 2.2. Patients and Ethics 

The analysis included 33 consecutive patients with mCRPC, who from 01/2016 to 10/2020, received ^177^Lu-PSMA-617 RLT as palliation given on a compassionate use basis under the German Pharmaceutical Act §13 (2b). These patients were included in our report regarding the first 254 men receiving ^177^Lu-PSMA-617 RLT in the REALITY study [9]. As summarized in Table 1, the present cohort comprised highly pretreated, mostly elderly men with late-stage to end-stage disease. A large majority of these patients had tumors at the most common sites of prostate cancer metastasis [25], the bones and lymph nodes, and tumors were present at less common sites of metastasis such as the liver in a relatively large proportion of this pre-selected group.

The inclusion criteria for this analysis were: histologically confirmed mCRPC; sufficient ^68^Ga-PSMA-11 PET/CT data to enable the calculation of TLP and SUVmean at baseline and post-RLT; and available data regarding molecular imaging response after the second ^177^Lu-PSMA-617 cycle. The exclusion criteria were: 18-fluoride-fluorodeoxyglucose (^18^F-FDG)-positive, PSMA-negative lesions, i.e., FDG/PSMA mismatch findings, if ^18^F-FDG PET/CT was performed, and prior 225-actinium-PSMA-617 RLT. Additionally, to avoid potential alteration of PSMA expression due to changes in the regimen of androgen deprivation therapy and/or novel androgen axis drugs [26], patients with such changes between baseline and post-RLT were ineligible for the analysis.

The REALITY study protocol was approved by the local Institutional Review Board, Ärztekammer des Saarlandes/Saarbrücken (permission number 140/17). After being thoroughly informed about the risks and potential adverse effects of ^177^Lu-PSMA-617 RLT, the patients gave written consent for such treatment; they also gave written consent for the related biochemical testing and ^68^Ga-PSMA-11 PET/CT, as well as the use of their de-identified data in the REALITY registry and in scientific publications.

### 2.3. ^68^Ga-PSMA-11 PET/CT

Each patient underwent ^68^Ga-PSMA-11 PET/CT 2.1 ± 2.1 weeks before the first cycle and 4.9 ± 2.0 weeks after the second cycle of ^177^Lu-PSMA-617 RLT; the median (minimum–maximum) interval between scans was 11 (5.9–19.3) weeks. In aggregate, patients received a median (minimum–maximum) 125 (77–166) MBq of ^68^Ga-PSMA-11; the median administered activities did not differ significantly between the two ^68^Ga-PSMA-11 PET/CT scans (median [minimum–maximum] 122 [77–166] MBq vs. 121 [100–147] MBq, *p* = 0.932). ^68^Ga-PSMA-11 was administered via intravenous infusion, which was followed by a 500 mL infusion of 0.9% NaCl. The interval from injection to imaging was ~60 min, per standard ^68^Ga-PSMA-11 PET procedures [27]. 

PET/CT scans were performed on a Biograph 40 mCT PET/CT scanner (Siemens Healthineers, Erlangen, Germany). The acquisition time was 3 min/bed position, there was an extended field-of-view of 21.4 cm (TrueV), and the slice thickness was 3.0 mm. For attenuation correction and anatomical localization, low-dose CT was acquired using an X-ray tube voltage of 120 keV and a modulation of the tube current applying CARE Dose4D software (Siemens Healthineers, Erlangen, Germany) with a reference tube current of 30 mAs. PET reconstruction was performed iteratively using a three-dimensional ordered-subset expectation maximization algorithm with three iterations, 24 subsets, Gaussian filtering, and a 5.0 mm slice thickness. As well as attenuation correction, decay correction, random correction, and scatter correction were applied.

### 2.4. Calculation of TLP

TLP, defined as the summed products of volume x uptake (as reflected by SUVmean) of all tumor lesions [28], was calculated employing a semiautomatic tumor segmentation algorithm based on that of Ferdinandus et al. [29], with a threshold of SUV ≥ 3 (Figure 1). Sites of probable ^68^Ga-PSMA-11 physiological uptake with an SUV above this threshold were manually excluded. Syngo.Via Enterprise VB 60 software (Siemens, Erlangen, Germany) was used.

### 2.5. ^177^Lu-PSMA-617 RLT

^177^Lu-PSMA-617 RLT regimens were individualized based on tumor burden and sites, the rate of tumor progression, bone marrow and renal function, and body surface area. The median (minimum–maximum) cumulative ^177^Lu-PSMA-617 activity of the two cycles of RLT was 12.6 (9.4–16.9) GBq. The median (minimum–maximum) administered activity was 6.9 (4.6–9.7) GBq for the first cycle and 6.2 (4.3–7.7) GBq for the second (*p* < 0.001). The mean interval between cycles was 5 ± 2 weeks.

### 2.6. Selection of Responders to ^177^Lu-PSMA-617 RLT

Subgroup selection for the comparison of SUVmeans in organs-at-risk in responders to RLT (*n* = 25/33, 76%) was based on the molecular imaging response, reflected by changes in TLP, and defined according to the modified PET Response in Solid Tumor Criteria (PERCIST) 1.0 [30]. Per those criteria, the threshold for partial response (PR) was defined as a TLP decline from baseline >30%.

### 2.7. Statistics

Data on the patients, imaging, and treatment characteristics are reported as descriptive statistics where applicable. Correlations between ∆TLP and ∆SUVmean and between baseline TLP and baseline SUVmean were evaluated using the Spearman’s correlation coefficient with 2-tailed testing for significance. Differences between the first and second scan and between RLT administered activities were compared via the Mann–Whitney *U* test, and for the SUVmean before and after RLT, differences were compared via the Wilcoxon signed-rank test.

To identify predictors of the tumor sink effects detected in the spleen and parotid gland, univariate regression was performed post hoc to determine the relationship, if any, between each of those two endpoints and each of five baseline characteristics and one RLT characteristic, analyzed as categorical variables. The characteristics analyzed (threshold for dichotimization) were: age (>75 years), Eastern Cooperative Oncology Group (ECOG) performance status (>2), TLP (>500 mL × SUV), prostate-specific antigen (PSA) level (>200 ng/mL), prior chemotherapy (present), and cumulative ^177^Lu-PSMA-617 activity after two cycles of RLT (>12.5 GBq). Variables with *p* ≤ 0.1 in the univariate analysis were to be included in a multivariable model to identify factors that independently predicted the tumor sink effect in each organ.

Excel (Microsoft, Seattle, WA, USA) and Prism version 8 (GraphPad Software, San Diego, CA, USA) were used for the statistical analyses. *p* values < 0.05 were considered to be statistically significant. 

## 3. Results

In the overall cohort (N = 33), ∆TLP and ∆SUVmean showed significant moderate inverse correlations in the parotid glands (r = −0.396, *p* = 0.023) and spleen (r = −0.356, *p* = 0.042) and a possible tendency toward such a correlation in the liver (r = −0.300, *p* = 0.089) (Figure 2). Analogously, molecular imaging (partial) responders to RLT (*n* = 25) had a significant increase from baseline to post-RLT in the organ SUVmean of the parotid glands (6.7 ± 2.1 vs. 7.6 ± 2.5, *p* = 0.022) and the spleen (5.1 ± 2.5 vs. 5.8 ± 2.6, *p* = 0.04) (Figure 3; representative image in Figure 4). The median (minimum–maximum) relative change in this variable was +12% (−40–+77%) in the parotid glands and +14% (−24–+111%) in the spleen. In the overall cohort (N = 33), the baseline TLP and baseline SUVmean exhibited significant moderate negative correlations in those same organs (parotid gland: r = −0.440, *p* = 0.01 and spleen: r = −0.418, *p* = 0.016) and trended towards significant, moderate negative correlation in the liver (r = −0.343, *p* = 0.051) (Figure 5). Regarding the kidneys, we observed no correlation between the ∆TLP and ∆SUVmean (r = 0.084, *p* = 0.646) (Figure 2), no significant increase from baseline to post-RLT in the SUVmean in RLT molecular imaging responders (22.9 ± 9.2 vs. 23.4 ± 9.5, *p* = 0.989; Figure 3), and no correlation between the baseline TLP and baseline SUVmean (r = −0.171, *p* = 0.351) (Figure 5).

No tested characteristic showed any significant univariate association with the tumor sink effect (Table 2). Because only one association, that of the baseline TLP and the tumor sink effect in the spleen, met the pre-specified threshold *p*-value for inclusion, multivariable analysis was not performed.

## 4. Discussion

This study is notable in taking a novel approach in assessing tumor sink effects with PSMA-targeted radiopharmaceuticals in men with prostate cancer. Namely, in the intra-individual comparisons involving 33 patients with mCRPC, we evaluated the relationship of *changes* in tumor burden effected by ^177^Lu-PSMA-617 RLT with *changes* in radiopharmaceutical uptake in key organs-at-risk in patients receiving such treatment. Additionally, in a second assessment, we focused on molecular imaging responders after two courses of ^177^Lu-PSMA-617 (*n* = 25), a subgroup in whom tumor sink effects would perhaps be likeliest to manifest. In these patients, we compared uptake in the same organs before and after RLT. Lastly, we also took a “traditional static approach”, performing intra-individual correlation of pre-RLT tumor burden and pre-RLT organ uptake in our entire cohort (N = 33).

Our key finding was that all three of these assessments suggested tumor sink effects of PSMA-targeted radiopharmaceuticals in the salivary glands and the spleen. The evidence comprised a significant moderate inverse correlation between ∆TLP and ∆SUVmean in these tissues, significant increases from baseline in median SUVmean in these organs in RLT responders, and a significant moderate negative correlation between the pre-RLT TLP and pre-RLT organ SUVmean. These observations closely align with findings in one or both of the salivary glands and splenic tissue in all of the five clinical studies [2,14,15,16,19], one simulation study [1], and one case report [20] of tumor sink effects in men with relatively advanced prostate cancer, including mCRPC (Table 3). These investigations used a variety of measures of tumor burden (e.g., total tumor volume, total PSMA-positive tumor volume, and subjective visual assessment) and organ radiopharmaceutical uptake (e.g., SUVmean, absorbed dose, and subjective visual assessment).

Less closely aligned with observations of most [2,14,19], but not all [16], of the four previously published studies and of the single case report [20] examining the issue in patients with higher tumor burdens, our findings suggest only a possible trend towards a tumor sink effect of PSMA-targeted radiopharmaceuticals in the liver. Here, evidence of a tumor sink effect was limited to a possible tendency (*p* = 0.09) towards a correlation of ∆TLP and ∆SUVmean and to a clearer trend (*p* = 0.051) towards a correlation of baseline TLP and baseline organ SUVmean. The reason for the apparently at best weaker tumor sink effect in the liver remains open and speculative. A possible explanation would be that because of this organ’s large volume, there may be relatively greater inaccuracy in determining the SUVmean; therefore, to detect a significant increase in the SUVmean, a higher number of cases might be needed.

In contrast, unlike all other investigators examining the question in patients with advanced disease [1,2,14,15,16,19,20], we detected no evidence of a tumor sink effect in the kidneys. The reasons for this observation also remain open and speculative. One possible explanation for at least some of the discrepant observations might be that Gaertner et al. [14] and Gafita et al. [19] used different methodologies than ours to determine the tumor sink effect. They compared the SUVmean of the kidneys between different patients rather than intra-individually, and they estimated tumor burden purely visually (i.e., subjectively), whereas we calculated the TLP,. because of its objectivity, a presumably more accurate method.

Of the preceding observations, that with the greatest clinical relevance may be the apparent tumor sink effect in the salivary glands. One of the main side effects of concern with RLT with ^177^Lu-PSMA-617 is xerostomia, because of its potentially important impact on patients’ quality of life. Thus, additional evidence that larger, potentially more effective RLT activities can be safely and tolerably administered in patients with high tumor burden without affecting salivary gland function is reassuring.

After we detected an apparent tumor sink effect in the parotid glands and spleen, we conducted a post hoc statistical analysis to identify factors predicting those findings. None of the six variables that we examined, baseline TLP, cumulative ^177^Lu-PSMA-617 activity after 2 cycles of RLT, age, ECOG performance status, PSA, or a history of chemotherapy, showed univariate association with either outcome. Only the relationship of a single variable, TLP, with a single outcome, a tumor sink effect in the spleen, had a *p* value beneath the *p* ≤ 0.1 threshold for inclusion in the multivariable analysis, precluding such testing. These observations highlight the importance of PSMA ligand imaging in predicting the likelihood of tumor sink effects. However, the findings of the only other study [2] to report a similar analysis conflict with our observations. Unlike us, Tuncel et al. [2] found sufficient variables with univariate associations with the tumor sink effect that they could perform multivariable analysis. In that analysis, they identified the total lesion PSMA index (metabolic tumor volume × SUVmean; a variable identical to our TLP), baseline PSA, and baseline PSA velocity as significant independent predictors of the tumor sink effect. However, it is difficult to compare those findings and ours, since only 3 variables (of 13 studied by Tuncel et al.) coincided in both univariate analyses and since Tuncel et al. defined the tumor sink effect visually and as a “whole-body” phenomenon, whereas we defined the effect using the SUVmean and focusing on specific organs. 

The limitations of our work should be acknowledged. First, ours was a single-center study with a relatively small sample, potentially increasing variability and limiting generalizability. Second, our cohort did not contain the wide gamut of patients with prostate cancer currently receiving imaging and/or treatment with PSMA ligand radiopharmaceuticals. Since two prior studies had suggested the absence of clinically relevant, if any, tumor sink effects in patients with metastatic hormone-sensitive [18] or earlier-stage [17] prostate cancer, we felt that focusing on late-stage patients with heavy tumor burdens would be the most fruitful investigative strategy. Third, our study assessed the tumor sink effect using sample-wide measurement and intra-individual comparison of key PET variables; we did not attempt to identify, or to compare RLT toxicity, in subgroups showing tumor sink effects versus those not showing such effects. Adverse events that we judged to be possibly or definitely attributable to RLT were relatively infrequent (affecting 3–36% of patients) and mild (grade ≤2 severity in all cases; grade 2 severity in only 4 cases) and, for the most part, were not clearly related to the organs-at-risk studied, i.e., the side effects comprised hematological abnormalities or fatigue in all but one instance, a case of grade 1 xerostomia presumably associated with salivary gland injury. Lastly, as also was seen in the other published studies [1,2,14,15,16,17,18,19] or the case report [20] on the topic, and due to practical difficulties of doing so in a retrospective “real-world” context, our study did not consider certain factors potentially affecting the biodistribution of PSMA-targeted radiopharmaceuticals. These factors may include trapping and excretion in pathological and healthy tissues, renal function, body composition and surface area, patients’ hydration, the content of their recent food intake, and even the time of day [1,2,3,31,32]. 

## 5. Conclusions

Our observations in patients with advanced mCRPC suggest tumor sink effects in the salivary glands and spleen and possibly in the liver, with PSMA-targeted radiopharmaceuticals. These findings and aligned observations previously reported in the literature provide a rationale for the empirical use and clinical study of serial adjustment of ^177^Lu-PSMA-617 activities over the course of RLT in such patients to try to improve the efficacy/toxicity ratio of this treatment. 

## Figures and Tables

**Figure 1 cancers-15-02592-f001:**
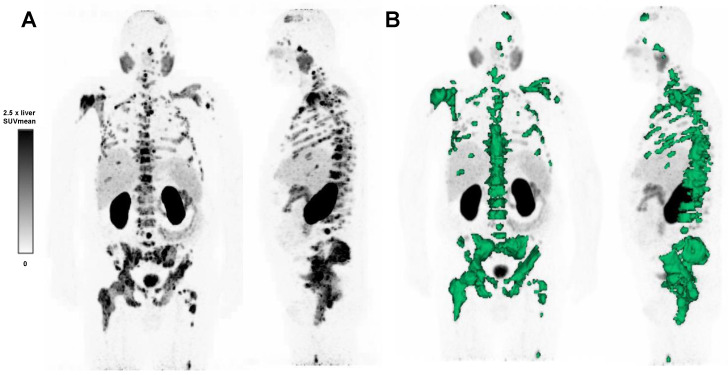
(**A**) ^68^Ga-PSMA-11 PET maximum intensity projection image of a patient with advanced mCRPC, and (**B**) semi-automatic tumor segmentation in that patient using Syngo.via Enterprise VB 60 (Siemens, Erlangen, Germany). The PSMA-positive tumor volume is delineated in green. The SUVmean window was set at 0–2.5 times the SUVmean of the healthy liver.

**Figure 2 cancers-15-02592-f002:**
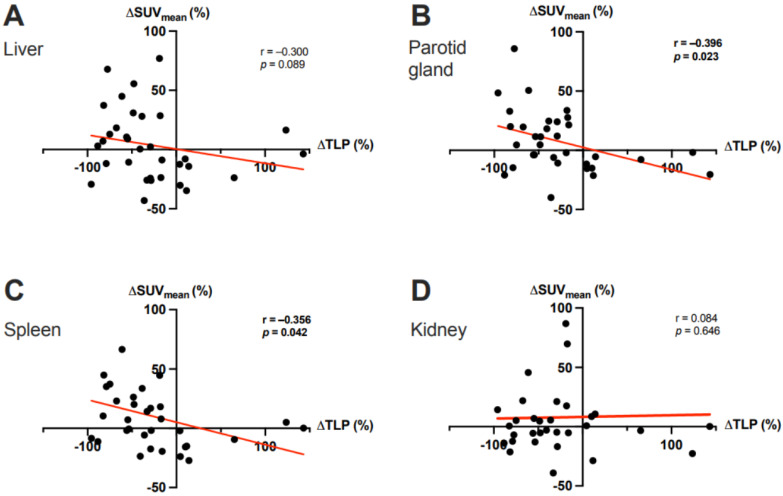
Correlation between the ΔTLP and the ΔSUV_mean_ in (**A**) the liver, (**B**) the parotid gland, (**C**) the spleen, and (**D**) the kidney after two cycles of ^177^Lu-PSMA-617 RLT. In panels **C** and **D**, 1 outlier each with a ΔSUV_mean_ >100% was cropped to simplify the presentation.

**Figure 3 cancers-15-02592-f003:**
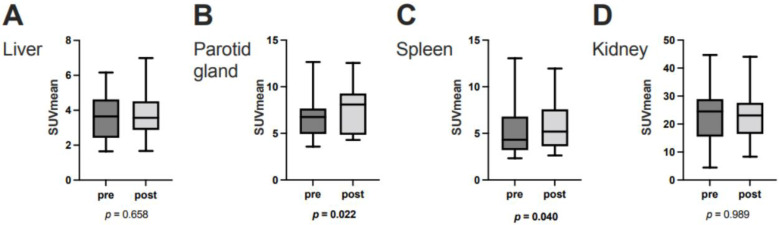
SUV_mean_ at baseline and after two cycles of ^177^Lu-PSMA-617 RLT in (**A**) the liver, (**B**) the right parotid gland, (**C**) the spleen, and (**D**) the right kidney in molecular imaging responders to PSMA-RLT (*n* = 25). Horizontal lines represent the respective median values.

**Figure 4 cancers-15-02592-f004:**
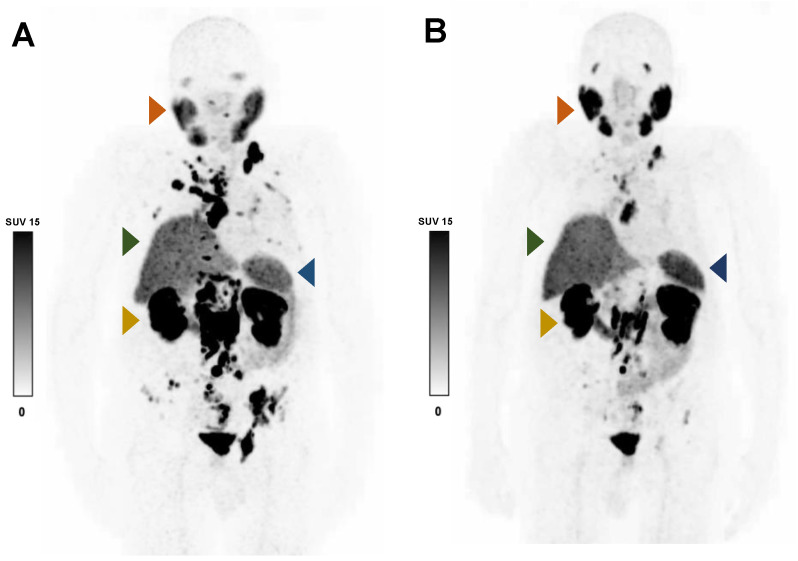
Representative ^68^Ga-PSMA-11 PET MIP images of an 86-year-old patient with advanced mCRPC who responded to RLT before (**A**) and after (**B**) two cycles of ^177^Lu-PSMA-617. After decreases in TLP (−83%) and serum PSA (−92%), the SUV_mean_ increased in the right parotid gland (ΔSUV_mean_ +32.9%; orange arrow) and the spleen (ΔSUV_mean_ +21.2%; blue arrow), but it was minimally changed in the liver (ΔSUV_mean_ +7.0%; green arrow) and right kidney (ΔSUV_mean_ +0.6%; yellow arrow).

**Figure 5 cancers-15-02592-f005:**
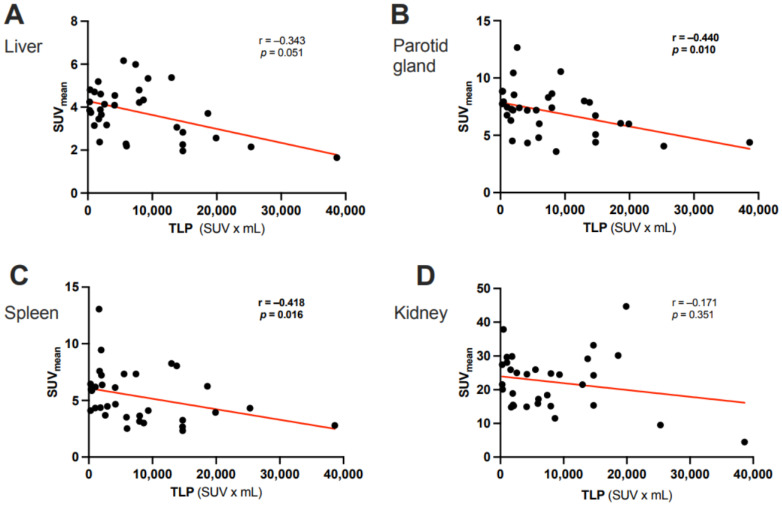
Correlation between the baseline TLP and the baseline SUV_mean_ of (**A**) the liver, (**B**) the parotid gland, (**C**) the spleen, and (**D**) the kidney in 33 patients with mCRPC.

**Table 1 cancers-15-02592-t001:** Patient characteristics (N = 33).

Patient Characteristics	Value
**Age**	
Median (minimum–maximum), yrs	72.5 (53–88)
Age ≥ 75 yrs, % (*n*)	42% (14)
**PSA** [ng/mL]	
Median (minimum–maximum)	208 (21–3025)
**ALP** [U/L]	
Median (minimum–maximum)	133 (35–1753)
**Hemoglobin** [g/dL]	
Median (minimum–maximum)	12 (6–16)
<13 g/dL, % (*n*)	48% (16)
**ECOG performance status**, % (*n*)	
0	21% (7)
1	36% (12)
≥2	43% (14)
**Sites of metastasis ^a^**, % (*n*)	
Bone	88% (29)
Lymph node	67% (22)
Liver	12% (4)
Other	18% (6)
**Prior therapies ^a^**, % (*n*)	
Radical prostatectomy	45% (15)
Radiation	58% (19)
ADT	100% (33)
Any NAAD	88% (29)
Abiraterone	73% (24)
Enzalutamide	61% (20)
Abiraterone and enzalutamide	45% (15)
Any chemotherapy	73% (24)
Docetaxel	73% (24)
Cabazitaxel	21% (7)
Docetaxel and cabazitaxel	21% (7)
[^223^Ra]Ra-dichloride	21% (7)
Other	15% (5)

ADT, androgen deprivation therapy; ALP, alkaline phosphatase; ECOG, Eastern Cooperative Oncology Group; NAAD, novel androgen axis drugs; PSA, prostate-specific antigen. ^a^ Patients could hjave received more than one of these therapies.

**Table 2 cancers-15-02592-t002:** Univariate regression analysis of potential predictive factors for the tumor sink effect in the spleen and right parotid gland.

Variable	Value	Spleen	Right Parotid Gland
	*% (n)*	*p*	*p*
**N** **Age ^a^**	100% (33)	-	-
≤75 years	58% (19)		
>75 years	42% (14)	0.957	0.321
**PSA ^a^**			
≤200 ng/mL	45% (15)		
>200 ng/mL	55% (18)	0.215	0.957
**Performance status ^a^**			
ECOG 0–1	58% (19)		
ECOG 2–3	42% (14)	0.123	0.255
**Cumulative ^177^Lu-PSMA-617 activity: first two cycles**			
≤12.5 GBq	48% (16)		
>12.5 GBq	52% (17)	0.873	0.873
**TLP ^a^**			
≤500 mL × SUV	48% (16)		
>500 mL × SUV	52% (17)	0.063	0.217
**Prior chemotherapy ^a^**			
No	27% (9)		
Yes	73% (24)	0.592	0.921

Because of rounding, the percentages may not add up to 100%. PSA, prostate-specific antigen; PSMA, prostate-specific membrane antigen; ECOG, Eastern Cooperative Oncology Group; TLP, total lesion PSMA. ^a^ Baseline variable.

**Table 3 cancers-15-02592-t003:** Evidence regarding the tumor sink effect with PSMA ligand radiopharmaceuticals: summary of the literature.

Study (N)	Design	Main Healthy-Organ-Related Endpoint(s)	Healthy Organ					
			Salivary Glands	Spleen	Liver	Kidney	Lacrimal Glands	Red Marrow
Gaertner et al., 2017 [14] (N = 135)	Retrospective inter-patient comparison ^a^	SUVmean	√	√	√	√	√	NS
Filss et al. 2018 [15] (N = 11)	Retrospective inter-patient evaluation ^a^	Kidney dose from one course of ^177^Lu-PSMA-617 RLT	√	NS	NS	√	NS	NS
Begum et al., 2018 [1] (N = 13)	Simulation study using a physiologically-based pharmacokinetic model to analyze actual patient data ^a^	Biologically-effective doses to the kidneys, salivary glands, and red marrow under simulated PSMA+ total tumor volumes of 0.1–10L	√	Non-significant	Non- significant	√	NS	The dose increased along with the total PSMA-positive tumor volume; this observation was attributed to higher whole-body retention with greater tumor burden
Violet et al., 2019 [16] (N = 30)	Prospective inter-patient evaluation ^a^	Mean absorbed dose	√	Non-significant	Non-significant	√	NS	Non-significant
Werner et al., 2020 [17] (N = 40)	Retrospective evaluation ^b^	Spearman’s rank correlation between tumor volume and organ uptake corrected to lean body mass or body weight	Non-significant	Non-significant	Non-significant	Non-significant	Non-significant	NS
Cysouw et al., 2020 [20] (N = 1)	Case report of one patient ^a^	Visual uptake in organs	√	√	√	√	NS	NS
Tuncel et al., 2021 [2] (N = 65)	Retrospective inter-patient comparison ^a^	Correlation of the SUVmax in the tumor and healthy tissue with the metabolic tumor volume (sum of volumes of tissue suspicious for malignancy with increased PSMA uptake) and the total lesion PSMA index (metabolic tumor volume x SUVmean)	√ ^d^	NS	√ ^d^	√ ^d^	NS	NS
Peters et al., 2022 [18] (N = 10)	Prospective dosimetry study embedded in a prospective clinical study ^c^	Correlation of the SUVmax in the tumor and healthy tissue with the metabolic tumor volume (sum of volumes of tissue suspicious for malignancy with increased PSMA uptake) and the total lesion PSMA index (metabolic tumor volume x SUVmean)	Non-significant	Non-significant	Non-significant	Non-significant	NS	NS
Gafita et al., 2022 [19] (N = 356)	International, multicenter retrospective analysis with inter-patient comparison ^a^	Correlation of the total PSMA-positive tumor volume as a continuous variable and by quintiles with organ SUVmean	√	√	√	√	NS	NS
Present study (N = 33, *n* = 25 for 1 comparison)	Retrospective intra-individual comparison ^a^	Correlation of ∆TLP and organ ∆SUVmean after two courses of ^177^Lu-PSMA-617; correlation of the baseline and post-RLT organ SUVmean in RLT molecular imaging responders; correlation of the baseline TLP and the baseline organ SUVmean	√	√	Possible trend or trend towards significance	Non-significant	NS	NS

√, tumor sink effect found; ^68^Ga, 68-gallium; ^177^Lu, 177-lutetium; CT, computed tomography; mCRPC, metastatic castration-resistant prostate cancer; NS, not studied; PET, postron emission tomography; PSMA, prostate-specific membrane antigen; RLT, radioligand therapy; SUVmax, maximum standardized uptake value; SUVmean, mean standardized uptake value; TLP, total lesion PSMA. ^a^ Studied predominantly or exclusively patients with advanced disease, including mCRPC. ^b^ Studied predominantly men with earlier-stage disease. ^c^ Studied men with metastatic hormone-sensitive disease. ^d^ The investigators found the tumor sink effect in 17/65 (26%) of the patients using a definition of (1) recovery (>30% increased tumor:background ratio) of decreased physiological hepatic, renal, and parotid gland uptake after two RLT cycles or (2) absent/mild visual uptake in physiological organs with extensive visual radiotracer uptake in metastatic sites.

## Data Availability

The datasets used and analyzed during the present study are available from the corresponding author upon reasonable request.

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
