# Peer review of "Tumor Sink Effect with Prostate-Specific Membrane Antigen-Targeted Theranostics in Patients with Metastatic Castration-Resistant Prostate Cancer: Intra-Individual Evaluations"

_cancers, 2023, doi:10.3390/cancers15092592_

Round 1
Reviewer 1 Report
I have gone through the manuscript . Topic is indeed interesting but it needs major modifications. Authors have consistently discussed about metastatic prostate cancer no information as provided related to the target organs of cancer cells for secondary tumor growth.
It is also essential to mention how efficiently radiotherapy inhibited metastatic spread. Because radiation exposure also triggered the generation of genomic rearrangement in the prostate cancer cells which can further aggravate the situation in the context of treatment.
Although the team has described the information related to human studies but brief information about this "tumor sink effects" needs to be discussed in animal model studies. Better and rationally designed approaches will be helpful for the cure with notably reduced side effects and off-target effects.
Authors should properly explain the concept of "TUMOR SINK EFFECT". It has been inappropriately explained. This aspect is really interesting and needs to be clarified for specialist and non-specialist readers.
Author Response
Please see the attachment. Please note that the attachment contains point-by-point response to the comments of Reviewer 1.

Reviewer 2 Report
Thank you for the opportunity to review this interesting article.
In this study, the authors assessed the tumour sink effects of PSMA radioligands in 33 men with mCRPC after 2 cycles of 177-Lu-PSMA, with the aim of describing a pathway to a safe dose adjusted radioligand therapy.
Overall, this article is well written and has appropriate literature search to complement the discussion. The table 3 is somewhat difficult to read and should be reformatted.
Otherwise, it is a good article close to publication and describes a pathway to a safe dose adjustment of radioligand therapy in the future.
The study could have been improved by describing the toxicity in those with significant tumour sink effect in the cohort of 33 patients.
Kind regards.
Author Response
Please see the attachment. Please note that the attachment contains point-by-point responses to the comments of Reviewer 2.

Round 2
Reviewer 1 Report
Looks in good form now.